# Knowledge Graph Construction
# with R2RML and RML:
# An ETL System-based Overview

Julián Arenas-Guerrero[1], Mario Scrocca[2], Ana Iglesias-Molina[1],
Jhon Toledo[1], Luis Pozo-Gilo[1], Daniel Doña[1], Oscar Corcho[1], and
David Chaves-Fraga[1]

[1] Ontology Engineering Group, Universidad Politécnica de Madrid, Spain
{julian.arenas.guerrero,ja.toledo,ana.iglesiasm
luis.pozo,daniel.dona,oscar.corcho,david.chaves}@upm.es
[2] Cefriel – Politecnico di Milano, Italy
mario.scrocca@cefriel.com

**Abstract.** Knowledge graphs have proven to be a powerful technology to integrate and structure the myriad of data available nowadays. The semantic web community has actively worked on data integration systems, providing an important set of engines and mapping languages to facilitate the construction of knowledge graphs. Despite these important efforts, there is a lack of objective evaluations of the capabilities of these engines in terms of performance, scalability, and conformance with mapping specifications. In this work, we conduct such evaluation considering several R2RML and RML processors to identify their strengths and weaknesses. We (i) perform a qualitative analysis of the distinctive features of each engine, (ii) examine their conformance with the mapping language specification they support, and (iii) assess their performance and scalability using the GTFS-Madrid-Bench benchmark.

**Keywords:** Knowledge Graphs · RML · R2RML · GTFS-Madrid-Bench

## 1 Introduction

In recent years, knowledge graphs (KGs) have become one of the most widely used technologies in data integration reaching the top positions of the Gartner Hype Cycle for Artificial intelligence in 2020[3]. This popularity has resulted in open KGs like Wikidata [25] or YAGO [12], and in the adoption of this technology by major technology companies such as Facebook, Google, or eBay [19]. To construct KGs from non-RDF data sources, mapping languages allow practitioners to define the relationships between input data sources and ontologies in

[3] https://www.gartner.com/smarterwithgartner/2-megatrends-dominate-the-gartner-hype-cycle-for-artificial-intelligence-2020/

a declarative and maintainable manner [17]. Although there are several mapping languages in the state of the art (e.g., SPARQL-Generate [16] or ShExML [9]), there are two of them that stand out: R2RML [5], which is the W3C standard language for RDB2RDF mapping, and RML [8], which is a well-known extension of R2RML for data formats beyond relational databases (RDBs).

KGs can be constructed with [R2]RML-compliant engines that can implement two strategies: materialization or virtualization [20]. The former is the ETL approach that generates the entire KG (i.e., all the triples), while the latter generates results for SPARQL queries by translating them to the native query language of the input data source (e.g., SQL queries in the case of RDBs) [3,21]. Given the high number of engines available [7,13,3,23,22,10], it is easy for any practitioner to get lost in deciding which one best fits their use case. While there are comprehensive and structured evaluations for the virtualization approach [4,15] that ease the user's choice, there is a lack of such an evaluation for materialization engines.

In this paper, we evaluate knowledge graph construction (KGC) engines that implement the materialization approach and support the [R2]RML mapping languages. First, we identify the most relevant engines available and provide a qualitative analysis of their features. Next, we assess their conformance to the language specification considering the test cases defined by each language [11,2]. Finally, we test the engines using the GTFS-Madrid-Bench benchmark [4] to evaluate their performance in terms of execution time, memory used and the number of triples generated using different data source formats and sizes.

The remainder of the article is structured as follows. Section 2 describes the related work on KGC systems and existing work evaluating them. Section 3 shows a qualitative analysis of several declarative KG engines where we highlight their strengths and weaknesses. Section 4 presents the quantitative experimental evaluation, using the test cases of each mapping language and the GTFS-Madrid-Bench benchmark. Finally, Section 5 provides a set of relevant conclusions extracted from the present work and the future lines of research.

## 2  Related Work

The emergence of different engines tackling KGC fosters the definition of benchmarks to evaluate their capabilities. The Berlin SPARQL Benchmark (BSBM) [1], based on fabricated data considering the e-commerce domain, allows the comparison of SPARQL queries performance posed against triplestores but also virtual KGC engines. The Norwegian Petroleum Directorate Benchmark (NPDB) [15], relying on real relational data from the oil industry, focuses on the requirements for virtualization and defines different test cases considering the parameters that impact the performance of KGC engines (different sizes of the data sources, types of queries, mappings, etc.). The GTFS-Madrid-Bench benchmark [4] considers data from the Madrid subway network and, reusing ideas from NPDB, defines a comprehensive set of test cases analyzing multiple requirements and considering data formats beyond relational DBMSs, namely CSV, JSON, and XML.

Despite the growing number of engines relying on the ETL approach, the efforts available in the literature analyzing KGC engines are mainly confined to the virtualization approach over RDB [26].

Materialization engines have been tested by their respective authors using *ad-hoc* evaluations. In consequence, the extracted conclusions are often limited, and cannot be generalized to all engines. SPARQL-Generate [16] relies on the SPARQL syntax to define mappings from heterogeneous data sources to RDF and it has been compared with RMLMapper[4] considering CSV datasets of different sizes. The authors of RMLStreamer [10] evaluated their tool against the SPARQL-Generate engine using artificial CSV, XML and JSON datasets and different sizes for these sources. The evaluation of RocketRML [23] considered a custom testbed using real touristic data on accommodations in JSON and XML formats, and was compared to RMLMapper. SDM-RDFizer [13] has been evaluated against RMLMapper and RocketRML using custom CSV datasets and considering different parameters beyond the data size, such as the factor of duplicates in the input data or different typologies of mappings. Finally, FunMap [14] defined a tabular-based testbed to assess the impact of transformation functions [6] in RML mappings, and the capability of their proposal executing functions in the initial phase of the KGC process. To the best of our knowledge, no comprehensive and structured evaluation has been carried out to assess the performance and scalability of materialization engines.

## 3  Knowledge Graph Construction Engines

In this section, we introduce the description and qualitative analysis of knowledge graph construction engines that implement the materialization approach. In the analysis, we limit ourselves to open-source [R2]RML systems. We assume that the reader is familiar with R2RML and RML. Table 1 shows a summary of the most relevant features of the selected engines.

### 3.1  R2RML Engines

The selected R2RML engines that we take into account are Ontop, Morph-RDB, db2triples, and R2RML-F. We excluded r2rml4net[5] as it only supports SQL Server (we use MySQL as the underlying DBMS for RDB in our evaluation). For each tool, we describe its main features and the strategies that they implement to improve the knowledge graph construction process.

**Ontop [3].** Ontop is a virtual knowledge graph system over relational databases developed at the Free University of Bozen-Bolzano and also supported by Ontopic s.r.l.[6]. Ontop focuses on translating SPARQL queries to SQL (i.e. virtualization). Nevertheless, the engine also has a materialization mode, which allows retrieving all the triples in a relational database. In order to achieve this, Ontop

---

[4] https://github.com/RMLio/rmlmapper-java

[5] https://github.com/r2rml4net/r2rml4net

[6] https://ontopic.biz/

Table 1: Summary of the features of knowledge graph construction systems.

| | Ontop | Morph-RDB | db2triples | R2RML-F | SDM-RDFizer | RMLMapper | Chimera | CARML | RocketRML | RML-streamer |
|---|---|---|---|---|---|---|---|---|---|---|
| **Data formats** | RDB | RDB | RDB | RDB, CSV | RDB, CSV, JSON, XML | RDB, CSV, JSON, XML | CSV, JSON, XML | CSV, JSON, XML | CSV, JSON, XML | CSV, JSON, XML |
| **Relational DBMS** | PostgreSQL, MySQL, SQL Server, Oracle, Db2 | Oracle, MySQL, H2 | PostgreSQL, MySQL, SQL Server, Oracle, MariaDB and others | PostgreSQL, MySQL, and Oracle | MySQL, PostgreSQL | MySQL, PostgreSQL, Oracle, and SQL Server | - | - | - | - |
| **Input data sources** | RDB Instance | RDB Instance | RDB Instance | RDB Instance, File | RDB Instance, File | RDB Instance, File, SPARQL endpoints | File, Named Streams | File, Named Streams | File | File, Named Streams |
| **Mapping languages** | R2RML, Ontop mappings | R2RML | R2RML | R2RML | RML | R2RML, RML, CSVW | RML | RML | RML | RML |
| **Functions** | no | no | no | yes | yes | yes | yes | yes | yes | no |
| **Output formats** | rdf/xml, turtle, n-triples, n-quads, trig | rdf/xml, rdf/xml-abbrev, n-triples, turtle, n3 | RDF/XML, n3, n-triples, turtle | n-triples, rdf/xml, rdf/xml-abbrev, n3, rdf/json, json-ld, n-quads, trig | n-triples, n-quads | n-quads, turtle, trig, trix, json-ld, hdt | binary, json-ld, n3, n-quads, n-triples, rdf/xml, turtle, rdfa | RDF4J Model | n-triples, n-quads, json-ld | n-quads, json-ld |
| **Named graphs** | yes | yes | no | yes | yes | yes | yes | yes | yes | yes |
| **Integrates multiple data sources** | no | no | no | Integrates different CSV files. | yes | yes | yes | yes | yes | yes |
| **Triplestore output** | no | no | no | no | no | no | yes | no | no | no |
| **Ontology input** | yes | no | no | no | no | no | yes (RDFS inference) | no | no | no |
| **Data errors** | no | no | no | no | no | no | no | no | no | no |
| **Chunk processing** | yes | no | no | no | no | no | no | no | no | no |
| **Implementation language** | java | scala | java | java | python | java | java | java | node | java |
| **Incremental writing results** | yes | yes | no | no | yes | no | yes | no | no | yes |
| **Duplicate removal** | yes | partially | yes | yes | yes | yes | yes | yes | yes | no |

creates a set of SPARQL queries that together generate all the triples of the knowledge graph. These SPARQL queries are then translated to SQL using the query translation capabilities of the system and are later executed against the underlying DBMS. The main advantage of this strategy is that Ontop generates efficient SQL queries thanks to the implementation of several optimizations. The engine is Java-based and supports the main DBMSs and RDF serializations. In addition, it allows processing SQL queries by chunks, avoiding the retrieval of large result sets at once.

**Morph-RDB [21].** Written in Scala, it is a virtual knowledge graph engine over relational databases developed at Universidad Politécnica de Madrid. Similarly to Ontop, it implements several query optimizations and provides a materialization mode. However, Morph-RDB does not generate SPARQL queries like Ontop, but it directly builds the necessary SQL queries for materialization. Namely, the engine generates one SQL query per triples map. If a triples map is composed of multiple referencing object maps, a join condition will be added to the SQL query per each of them. Hence, complex SQL queries with many join conditions may be generated. During materialization, Morph-RDB does not apply any of the optimization implemented for the virtualization mode. This results in queries that cannot be efficiently processed by DBMSs in some cases. Triples are written to a file using N-Triples format. Other serialization formats are supported by delegating to Apache Jena. Materialization of relational sources performed by the Morph-xR2RML [18] engine works similarly to Morph-RDB, but Morph-xR2RML additionally supports the materialization of NoSQL databases and processes the xR2RML mapping language [18], an R2RML extension.

**db2triples**[7]**.** db2triples is a Java materialization engine developed by Antidot[8]. The materialization is done using the algorithm proposed in the R2RML specification[9]. This algorithm independently executes each referencing predicate object map in a triples map, reducing the number of join conditions in the generated SQL queries w.r.t. Morph-RDB. As far as we know, the engine does not implement materialization optimizations.

**R2RML-F [7].** R2RML-F is an extension of db2triples developed at Trinity College Dublin. It allows executing additional data transformations encoded in the mappings via functions. Additionally, R2RML-F further extends db2triples to support named graphs and CSV data sources. In order to process CSV sources, these are previously loaded into an in-memory RDBMS, creating a native RDB schema using the column names of the CSV files.

### 3.2 RML Engines

The selected RML engines that we take into account for this analysis are RMLMapper, CARML, RMLStreamer, SDM-RDFizer, RocketRML, and Chimera. The selection criteria is the RML implementation report[10], which is up-to-date, in

---

[7] https://github.com/antidot/db2triples

[8] https://www.antidot.net/

[9] https://www.w3.org/2001/sw/rdb2rdf/r2rml/#generated-triples

[10] https://rml.io/implementation-report/

contrast to the R2RML one. Similarly to R2RML engines, for each tool we provide a description of its main characteristics together with the approaches implemented to make the materialization process more efficient.

**RMLMapper**[11]**.** RMLMapper is the reference implementation for an RML processor. It aims at being a feature-complete engine and ensuring compliance with the RML specification. The system, developed at Ghent University, is an in-memory Java-based processor that also supports R2RML mappings and can process different local and remote data sources (CSV/JSON/XML files, RDBs, SPARQL endpoints). RMLMapper supports the integration of RML and transformation functions, defining mechanisms to preload or dynamically load at runtime the functions referenced in the RML mappings. A set of in-memory caches are implemented to avoid multiple parsing procedures on the same input data source and multiple executions of the same triples map.

**SDM-RDFizer** [13]**.** Written in Python and developed by TIB Leibniz Information Center for Science and Technology, SDM-RDFizer provides a set of physical data structures for the efficient construction of the knowledge graph. More in detail, SDM-RDFizer provides two different structures: i) the Predicate Tuple Table that stores the triples associated with each predicate and, ii) the Predicate Join Tuple Table that stores the values of the subjects generated by a triples map that are involved in simple and complex referencing predicate object maps. Associated with these two structures, SDM-RDFizer also implements efficient physical operators to manage those tables. These structures and operators are focused on efficiently removing duplicated triples and contributing to improve the performance of join conditions.

**RocketRML** [23]**.** RocketRML is a Node-based RML parser developed by STI Innsbruck that supports CSV, JSON, and XML data sources. The engine introduces optimizations over join conditions for improving their execution performance. Before starting the mapping process for a triples map, the implementation checks whether it is in the join condition of another triples map. If it is, the parent triples map of the join condition is evaluated and the obtained values are cached. Then, all triples maps are parsed iteratively and at the end of the execution, the cached hash table is used to generate the triples associated with rules with join conditions. RocketRML constructs an in-memory knowledge graph and its principal output format is JSON-LD. When N-Triples is required, it delegates the removal of duplicates to an external library.

**CARML**[12]**.** CARML, currently developed by Skemu[13], is an engine supporting CSV, JSON, and XML data sources. The system is available as a Java library, and it relies on the RDF4J library generating an RDF4J Model as a result of the mapping processing. CARML introduces two extensions (prefix `carml:`) to the RML specification in order to support: (i) named streams as input data sources for the transformation (`carml:Stream`), (ii) specification of XML namespaces used in XPath expression for XML data sources (`carml:declaresNamespace`).

---

[11] `https://github.com/RMLio/rmlmapper-java`
[12] `https://github.com/carml/carml`
[13] `https://skemu.com`

Moreover, the engine defines a simplified mechanism based on Java annotations to bind the implementation of transformation functions. CARML defines an extensible mechanism to easily change the implementation of the logical source resolvers to access the input data sources. In contrast to other engines, such as RMLMapper, it optimizes the parsing procedure by adopting different Java libraries to access CSV, JSON and XML data sources.

**RMLStreamer [10].** RMLStreamer is a system developed at Ghent University that incorporates parallelization in the RDF generation process. Implemented in Scala, it is built on top of the distributed processing framework Apache Flink. RMLStreamer considers three main tasks: i) ingestion: implemented as an input operator on a Flink channel that depending on the data source can use a parallel or sequential input operator, ii) mapping processor: implemented as a transformation operator that reads data records from the ingestion task buffers, and generates RDF in its own buffer, iii) combination: it implements an output operator where it merges all the intermediate results in all the mapping processor buffers.

**Chimera [22].** Chimera is a framework developed by Cefriel[14]. It is implemented on top of Apache Camel and provides a set of building blocks to compose conversion pipelines based on Semantic Web solutions. A basic conversion pipeline is based on a lifting block to convert heterogeneous data sources into an RDF representation, and a lowering block to convert the obtained triples to the target data format. The default implementation of the lifting block is based on RMLMapper. To better support the Apache Camel framework, Chimera adds support for named streams as input data sources. Differently from CARML, it does not extend the RML specification, but it introduces an alternative and configurable access mechanism for the input data sources specified in the RML mappings. To improve the execution time, Chimera extends RMLMapper to define a multi-thread safe materialization procedure and introduces different options for concurrently processing the mappings. Moreover, it optimizes the parsing procedure for JSON and XML data sources. In order to improve memory consumption, the engine implements output writing to an external triplestore/SPARQL endpoint, and a mechanism for incremental/concurrent writes to optimize the processing of large datasets. Furthermore, it implements options to avoid the usage of caches when not needed.

## 4 Comparison Framework

In this section, we address the capabilities of the engines previously presented. First, we address the conformance of the engines w.r.t. the mapping languages specifications, and then we empirically analyze their performance. The versions of the engines used in this comparison are the following: Ontop v4.1.0, Morph-RDB v3.12.5, db2triples v2.2, R2RML-F v1.2.3, RMLMapper v4.9.1, CARML v0.3.1, RocketRML v1.8.2, SDM-RDFizer v3.5, RMLStreamer v2.0 and Chimera

---

[14] https://www.cefriel.com/

**Table 2:** R2RML conformance of state of the art engines. The results provide the number of test cases passed and failed.

|  | PostgreSQL | | MySQL | |
|---|---|---|---|---|
|  | passed | failed | passed | failed |
| Morph-RDB | 27 | 35 | 31 | 31 |
| Ontop | 59 | 3 | 45 | 17 |
| DB2Triples | 13 | 49 | 24 | 38 |
| R2RML-F | 13 | 49 | 24 | 38 |

**Table 3:** RML conformance of state of the art engines. The results provide the number of test cases passed and failed. Minus symbol denotes that the test-case is not applicable to an engine (the input data source is not supported).

|  | CSV | | JSON | | XML | | PostgreSQL | | MySQL | |
|---|---|---|---|---|---|---|---|---|---|---|
|  | passed | failed | passed | failed | passed | failed | passed | failed | passed | failed |
| RMLMapper | 38 | 1 | 38 | 2 | 36 | 2 | 53 | 7 | 50 | 10 |
| CARML | 25 | 14 | 23 | 17 | 23 | 15 | - | - | - | - |
| RocketRML | 29 | 10 | 30 | 10 | 29 | 9 | - | - | - | - |
| SDM-RDFizer | 24 | 15 | 23 | 17 | 21 | 17 | 11 | 49 | 20 | 40 |
| RMLStreamer | 39 | 0 | 40 | 0 | 38 | 0 | - | - | - | - |
| Chimera | 36 | 3 | 37 | 3 | 35 | 3 | - | - | - | - |

v2.1. The resources used for the comparison of the engines are available in a public repository[15].

### 4.1 Specification Conformance

To assess the conformance of the engines with their corresponding mapping language specifications, we rely on the test cases defined for R2RML [24] and RML [11]. These consist of a collection of tests that are used to check whether the engines support the requirements defined in the specifications. The results of these test cases also provide useful information for practitioners to select the engine that best fits their use cases. Additionally, they help developers to identify possible issues when parsing mapping rules.

**R2RML test cases.** Table 2 shows the number of R2RML tests passed and failed by the engines. Since R2RML-F is based on the code from db2triples, their results are similar. These two systems fail in most of the test cases because they do not support delimited identifiers in SQL. Although Morph-RDB usually

---

[15] https://github.com/oeg-upm/kgc-eval

outputs an error when it is expected by the test case, the engine also generates an empty graph which causes many of them to fail. Ontop performs well over PostgreSQL but it presents problems when the RDBMS is MySQL. Particularly, it fails when using `rr:graphMap`, and also in those test cases assessing the correctness of subject URIs.

**RML test cases.** Table 3 presents the results obtained by RML engines. RMLMapper, RMLStreamer, and Chimera cover most of the mapping language specification. The failed tests for RMLMapper are related to the automatic datatyping of literals in RDB [11]. Although RocketRML supports `rr:graphMap`, most of the reported failures occur when this property is included in the mapping rules. SDM-RDFizer does not pass some test cases because it does not support the generation of blank nodes. Additionally, in the same manner as Morph-RDB, SDM-RDFizer generates an empty graph when an error occurs. Finally, CARML failures are related to the possibility to define multiple subject maps, multiple predicate maps, and named graphs in the mapping rules.

## 4.2 Performance and Scalability with GTFS-Madrid-Bench

To test the performance and scalability of KGC engines, we rely on the GTFS-Madrid-Bench benchmark [4]. This benchmark provides a generator to create several distributions in different formats and sizes of the data.

**Datasets and Mappings.** Using the generator provided by the benchmark, we generate 5 different distributions based on the data format: $GTFS^{csv}$, $GTFS^{rdb}$, $GTFS^{xml}$, $GTFS^{json}$ and $GTFS^{custom}$. We also generate different data sizes of these distributions considering the scaling factors: $GTFS_1$, $GTFS_{10}$, $GTFS_{100}$, and $GTFS_{1000}$. We have used MySQL 8.0 as DBMS for $GTFS^{rdb}$. The benchmark already provides the mappings in [R2]RML languages. They are composed of 73 predicate object maps, of which, 12 are referencing predicate object maps. Most engines were not being able to process the referencing object map that joins the triples maps *shapes* and *shapePoints*. For this reason, we have transformed it into an equivalent predicate object map that does not affect the results.

**Engines.** The engines have been configured so that they do not generate duplicated triples and write the output RDF in N-Triples format. For Java engines, the heap size used for the different benchmark sizes has been configured accordingly to avoid heap errors. We have excluded RMLStreamer, which does not support removing duplicated triples, and we have opened an issue with this problematic in its public repository[16].

**Metrics.** We consider three metrics for the evaluation: execution time, maximum memory used and the number of triples generated. This last metric assesses the correctness of the generated RDF. However, it is limited, since it does not take into account aspects such as the generation of datatypes or whether data is extracted properly from the sources. A *timeout* of 24 hours is used. The experiments are executed in a CPU Intel(R) Xeon(R) Silver 4216 CPU @ 2.10GHz, 20 cores, 128 Gb RAM and a SSD SAS Read-Intensive 12 Gb/s.

---

[16] `https://github.com/RMLio/RMLStreamer/issues/26`

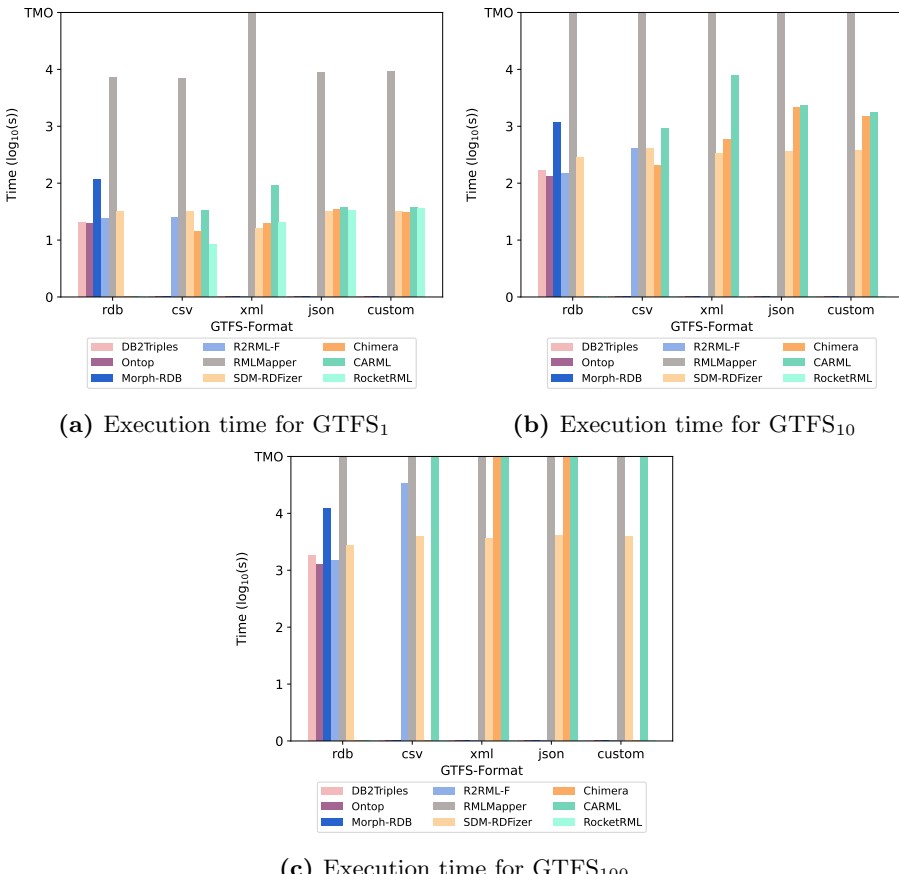

**(a)** Execution time for $\text{GTFS}_1$      **(b)** Execution time for $\text{GTFS}_{10}$

**(c)** Execution time for $\text{GTFS}_{100}$

**Fig. 1: Total execution time of GTFS-Madrid-Bench.** KGC time in seconds (logarithmic scale) for nine engines runned over $\text{GTFS}^{csv}$, $\text{GTFS}^{rdb}$, $\text{GTFS}^{xml}$, $\text{GTFS}^{json}$ and $\text{GTFS}^{custom}$ for three data size scaling factors: (a) $\text{GTFS}_1$, (b) $\text{GTFS}_{10}$ and (c) $\text{GTFS}_{100}$. The absence of the bar indicates an *out-of-memory* issue. The bars reaching the top means a *timeout* issue.

**Results.** Because of the fact that all engines resulted in *timeout* or *out-of-memory* for $\text{GTFS}_{1000}$ in all data formats, we have omitted this data scaling factor for the benchmark in the results. Figure 1 depicts the execution times obtained. For the case of $\text{GTFS}^{rdb}$, the execution times for db2triples, Ontop, R2RML-F, and SDM-RDFizer are in the same order of magnitude. Nonetheless, SDM-RDFizer needs more than twice as long as Ontop to generate all the results for $\text{GTFS}^{rdb}_{100}$, possibly due to SDM-RDFizer not pushing down the joins to the DBMS. The high number of joins in the SQL queries used by Morph-RDB results in high query execution times, and the inefficient duplicates elimination strategy of RMLMapper causes the engine to reach *timeout* for data sizes greater than

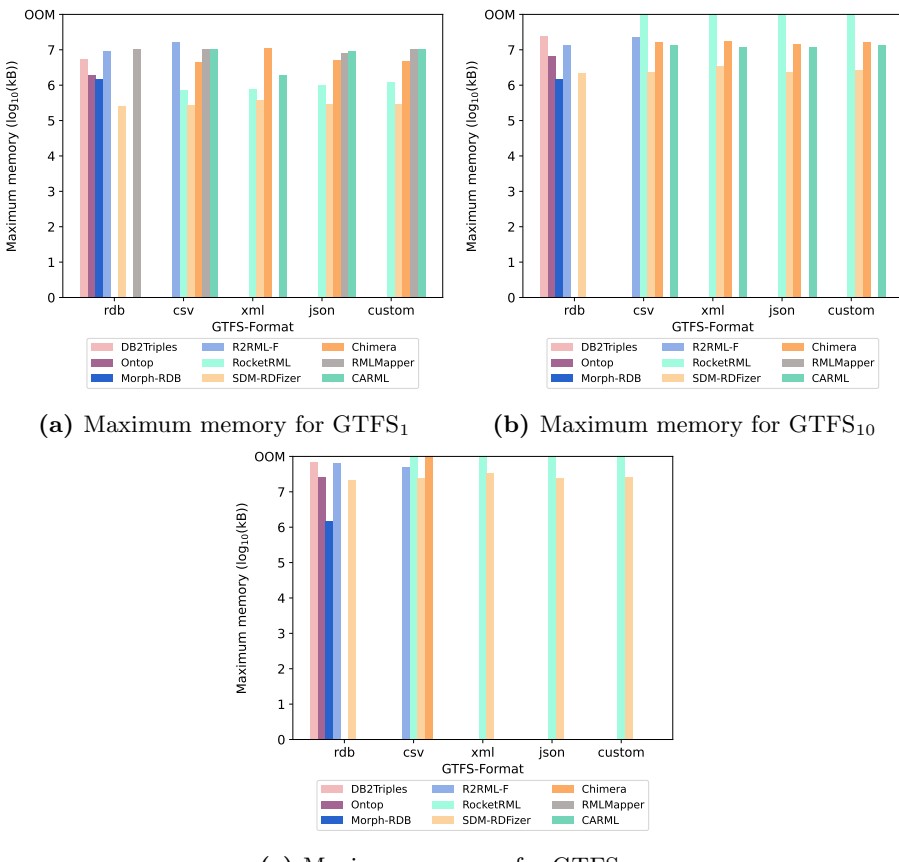

**(a)** Maximum memory for $GTFS_1$      **(b)** Maximum memory for $GTFS_{10}$

**(c)** Maximum memory for $GTFS_{100}$

**Fig. 2: Memory consumption peak of GTFS-Madrid-Bench.** KGC maximum memory consumption in kB (logarithmic scale) for nine engines runned over $GTFS^{csv}$, $GTFS^{rdb}$, $GTFS^{xml}$, $GTFS^{json}$ and $GTFS^{custom}$ for three data size scaling factors: (a) $GTFS_1$, (b) $GTFS_{10}$ and (c) $GTFS_{100}$. The absence of the bar indicates a *timeout* issue. The bars reaching the top means an *out-of-memory* issue.

$GTFS_1^{rdb}$. For $GTFS^{csv}$, Chimera manages to be the fastest tool for $GTFS_{10}^{csv}$ thanks to parallelization, but it runs out of memory for $GTFS_{100}^{csv}$. R2RML-F and SDM-RDFizer are the only engines capable of generating all the results for $GTFS_{100}^{csv}$, but the later is more than seven times faster than the former. For the rest of the data formats, SDM-RDFizer stands out, being the only engine capable of materializing $GTFS_{100}^{xml}$, $GTFS_{100}^{json}$, and $GTFS_{100}^{custom}$.

Maximum memory used by engines is presented in Figure 2. Since most engines keep the entire KG in memory to eliminate duplicated triples, the memory consumption is high. The exception is Morph-RDB that uses a different strat-

**Table 4:** Number of triples generated by each engine per data format and size for GTFS-Madrid-Bench benchmark. The absence of value means that the engine does not support the format and zero value means that the engine outputs a *timeout* or an *out-of-memory* error.

| | Ontop | Morph-RDB | db2triples | R2RML-F | SDM-RDFizer | RML Maper | Chimera | Rocket RML | CARML |
|---|---|---|---|---|---|---|---|---|---|
| **GTFS-1** | | | | | | | | | |
| **RDB** | 395953 | 454661 | 395953 | 395953 | 395953 | 397622 | - | - | - |
| **CSV** | - | - | - | 395953 | 395953 | 397622 | 395953 | 395953 | 395953 |
| **JSON** | - | - | - | - | 395953 | 397622 | 395953 | 397622 | 397622 |
| **XML** | - | - | - | - | 395953 | 397622 | 395953 | 395953 | 395953 |
| **CUSTOM** | - | - | - | - | 395953 | 397622 | 395953 | 395953 | 395953 |
| **GTFS-10** | | | | | | | | | |
| **RDB** | 3959530 | 4546610 | 3959530 | 3959530 | 3959530 | 0 | - | - | - |
| **CSV** | - | - | - | 3959530 | 3959530 | 0 | 3959530 | 0 | 3959530 |
| **JSON** | - | - | - | - | 3959530 | 0 | 3959530 | 0 | 3976220 |
| **XML** | - | - | - | - | 3959530 | 0 | 3959530 | 0 | 3959530 |
| **CUSTOM** | - | - | - | - | 3959530 | 0 | 3959530 | 0 | 3959530 |
| **GTFS-100** | | | | | | | | | |
| **RDB** | 39595300 | 45466100 | 39595300 | 39595300 | 39595300 | 0 | - | - | - |
| **CSV** | - | - | - | 39595300 | 39595300 | 0 | 0 | 0 | 0 |
| **JSON** | - | - | - | - | 39595300 | 0 | 0 | 0 | 0 |
| **XML** | - | - | - | - | 39595300 | 0 | 0 | 0 | 0 |
| **CUSTOM** | - | - | - | - | 39595300 | 0 | 0 | 0 | 0 |

egy for duplicate elimination. R2RML-F and db2triples are the ones with the highest memory use for $GTFS^{rdb}$, while Ontop and SDM-RDFizer require less than half of the memory. For the rest of data formats, SDM-RDFizer is the most memory-efficient engine as observed in the graphic of $GTFS_{10}$.

Table 4 shows the number of triples generated by the engines. For $GTFS_1$ it can be observed that the engines that do not generate the expected number of results are Morph-RDB, RMLMapper, and RocketRML and CARML for $GTFS^{json}$. Morph-RDB attempts to remove duplicates by using the *DISTINCT* clause in the SQL queries, which is not enough to remove all of them. RMLMapper, and RocketRML and CARML for $GTFS^{json}$ generate a higher number of triples because they generate triples for empty data values. In addition, RMLMapper generates triples in $GTFS^{rdb}$ when columns contain *NULLs*, which does not conform with R2RML specification.

## 5 Conclusions and Future Work

In this work, we address the problem of executing a comprehensive and structured evaluation over declarative knowledge graph construction engines that implement the materialization approach. We present a qualitative analysis of

the engines including their main features, optimizations, and limitations. We also carry out a quantitative evaluation of the engines by first assessing their conformance with the mapping language specification using their test cases, and then testing their performance using the GTFS-Madrid-Bench benchmark. The results obtained suggest that: i) there are few systems with high coverage of the features considered in our qualitative analysis; ii) several engines have a medium-low conformance w.r.t. the mapping languages specifications; and iii) most of the engines report performance and scalability problems for large input data sources.

After our experience in testing the capabilities of KGC engines, we devise a set of future working lines that can conduct the research and development of a new generation of systems. First, new generalized optimizations for the ETL approach are desirable to scale current systems to big data integration scenarios. Second, work on the conformance of the systems with the specifications and increase the efforts to have feature-rich systems. Finally, new versions of the current mapping language specifications will be required for more complex data integration problems (e.g., transformation functions) together with the extension of benchmarks to assess them.

### Acknowledgments

The work presented in this article is supported by the project Semantics for PerfoRmant and scalable INteroperability of multimodal Transport (SPRINT H2020-826172), by the Spanish Ministerio de Economía, Industria y Competitividad and EU FEDER funds under the DATOS 4.0: RETOS Y SOLUCIONES - UPM Spanish national project (TIN2016-78011-C4-4-R), and by an FPI grant (BES-2017-082511).

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
