# OpenReview forum: "Knowledge Graph Construction with R2RML and RML: An ETL System-based Overview"
_eswc-conferences.org/ESWC/2021/Workshop/KGCW — KGCW 2021_

### Official Review · ~Jakub_Klímek1 · 2021-03-31
**Handy comparison of (R2)RML based ETL tools with clear identification of their limitations**

**Rating:** 8
**Confidence:** 4

**Review:**

The authors present a clearly focused, defined, and executed comparison of ETL-based (R2)RML engines on a benchmark dataset.
Also, limitations and various incompatibilities are pointed out for each of those engines, providing the reader the necessary information for a decision about which of those engines to use in their project.

My only concern is with the title of the paper, which suggests that the focus of the paper is on all ETL software usable for KG generation, which is not true. The paper is actually focused only on declarative ETL systems which implement (R2)RML mappings. In many real-world KG generation cases, this is not required. Therefore, pointing this focus out in the paper title would seem appropriate.

Minor issue
- Conclusions: "and development of a new generating of systems." probably should be "generation of systems"

---

### Official Review · ~Manolis_Koubarakis1 · 2021-04-02
**Interesting paper surveying and comparing R2RML and RML processors.**

**Rating:** 7
**Confidence:** 5

**Review:**

The paper surveys and compares several R2RML and RML processors with the goal of identifying their
strengths and weaknesses.

The contributions of the paper are the following:
* A detailed qualitative analysis of the distinctive features of each system
* A study of the conformance of each system with the mapping language specification they support
* An evaluation of the performance and scalability of the studied systems using the GTFS-Madrid-Benchmark

The paper is well-written and easy to read.

---

### Official Review · ~Miel_Vander_Sande2 · 2021-04-14
**Interesting comparison between the main RML implementations**

**Rating:** 6
**Confidence:** 5

**Review:**

This paper evaluates a number of implementations of the RDB-to-RDF Mapping Language (R2RML) and its extension RDF Mapping Language (RML) using a GTFS-based benchmark. The paper is easy to read and well written overall. It's main flaw is its framing: this is not a study on ETL systems for Knowledge graph Construction, but a simple comparison survey on open-source (R2)RML implementations. As a result, the paper oversells its contributions, mostly in sections 1 and 3. Fortunately for the authors, this is easily fixed.

In the introduction, the authors position their work as a broad study on ETL approaches, but as we later find out, they restrict themselves to declarative languages, more specifically open-source systems that support R2RML and RML. Also, the authors talk about Knowledge Graphs, while they only mean RDF KG's. These are not the only kind of KGs.
The set of requirements in Section 3 that point this out are rather arbitrary and not properly motivated. Why not SPARQL-Generate, TARQL or XR2RML? Aren't those declarative as well? Why do the systems "need" to be open-source; there is freeware as well and did they authors try to contact companies like OpenLink? The requirements are clearly written to fit the interest of the authors, while the solution is much simpler: if the paper is about (R2)RML systems, then simply say so from the beginning. A paper should only talk about the things that were studied and make claims backed up by evidence. This would also result in a shorter, more to-the-point paper. Hence, I urge  the authors to completely revise Sections 1 - 3 to (only) reflect the work in Sections 3 & 4.

Other comments:
- Sec 1: RML is positioned as the de-facto standard because many tools implement it. Can you provide a citation for that?
- Sec 1/2: the paper is not consistent in describing what is going to be measured. It mentions "evaluation", "performance", then "performance and scalability"
- Sec 3.1: the argument to "exclude r2rml4net because SQL server is not in the the test cases" is a rather weak argument
- Table 1: mentions Function, but this hasn't been explained at all
- Section 4.2: How tailored is the GTFS test dataset to the SDM-RDFizer by the same authors? Related remark: The JSONPath/XPath coverage of SDM-RDFizer is unknown and at least partial. Wouldn't that bring implementations that focus on full compliance (eg. RMLMapper that uses a library) in a performance disadvantage? Why wasn't this considered when discussing the results?
- Section 4.2: PostgresSQL 12 also supports (parts of) JSONPath. This would be interesting to include Postgres JSON, Ontop and R2RML in future benchmarks
- Section 5: the conclusion is mostly summary and stating the obvious. What concrete steps can you propose given your results? I urge the authors to write a more compelling conclusion.

---

### Meta-Review · Program_Chairs · 2021-04-21

**Recommendation:** Accept
**Confidence:** 5

**Metareview:**

This paper provides an overview and comparison of (R2)RML open source processors. The reviewers agree that this is a well-written and easy-to-read paper. However, most reviewers point out that the paper in principle investigates only declarative ETL systems and it narrows down even more and covers the ones that implement (R2)RML mappings. The reviewers argue that this is not reflected in the title and introduction that go broader. We would thus advice the authors to scope the paper more accurately from the beginning making it more to the point in the first place and give an explanation about the requirements' choice because in their current form they seem to be arbitrarily chosen according to some of the reviewers.

---

### Decision · Program_Chairs · 2021-04-23

Accept